# Advances in Immunotherapy in Hepatocellular Carcinoma

**DOI:** 10.3390/ijms26051936

**Published:** 2025-02-24

**Authors:** Matthew Bloom, Sourav Podder, Hien Dang, Daniel Lin

**Affiliations:** 1Department of Medical Oncology, Sidney Kimmel Cancer Center, Thomas Jefferson University Hospital, Philadelphia, PA 19107, USA; matthew.bloom@jefferson.edu; 2Department of Surgery, Thomas Jefferson University Hospital, Philadelphia, PA 19107, USA; sourav.podder@jefferson.edu (S.P.); hien.dang@jefferson.edu (H.D.)

**Keywords:** hepatocellular carcinoma, immunotherapy, immune checkpoint inhibitors

## Abstract

Over the past several years, the therapeutic landscape for patients with advanced, unresectable, or metastatic hepatocellular carcinoma has been transformed by the incorporation of checkpoint inhibitor immunotherapy into the treatment paradigm. Frontline systemic treatment options have expanded beyond anti-angiogenic tyrosine kinase inhibitors, such as sorafenib, to a combination of immunotherapy approaches, including atezolizumab plus bevacizumab and durvalumab plus tremelimumab, both of which have demonstrated superior response and survival to sorafenib. Additionally, combination treatments with checkpoint inhibitors and tyrosine kinase inhibitors have been investigated with variable success. In this review, we discuss these advances in systemic treatment with immunotherapy, with a focus on understanding both the underlying biology and mechanism of these strategies and their efficacy outcomes in clinical trials. We also review challenges in identifying predictive biomarkers of treatments and discuss future directions with novel immunotherapy targets.

## 1. Introduction

Despite improvements in screening and treatment strategies, hepatocellular carcinoma (HCC) remains the third leading cause of cancer-related mortality in the world [1]. The prognosis of HCC is largely associated with the tumor stage. If diagnosed early, curative therapies including surgical resection, radiofrequency ablation, and liver transplantation, may achieve long-term survival. However, most patients are diagnosed at more advanced stages, where cure is no longer attainable, and treatment options, including locoregional therapy and systemic therapy, are considered palliative [2]. Therefore, there has been a strong need to improve therapeutic strategies for patients with advanced HCC.

In recent years, the treatment landscape of advanced, unresectable, or metastatic HCC has expanded significantly, from anti-angiogenic tyrosine kinase inhibitor (TKI) therapy options to the incorporation of immune checkpoint inhibitor (ICI) therapy. ICIs have demonstrated improved outcomes in advanced HCC and are integral to the overall treatment paradigm. ICIs targeting checkpoint proteins such as programmed cell death protein 1 (PD-1), programmed death-ligand 1 (PD-L1), and cytotoxic T-lymphocyte-associated protein-4 (CTLA-4) have demonstrated anti-tumor response in HCC [3,4,5,6]. Although early phase studies with single-agent anti-PD-1 inhibitors showed potential for durable tumor responses, subsequent phase III trials could not establish the superiority of anti-PD-1 monotherapy over the long-held standard of anti-angiogenic TKI therapy, such as sorafenib, in the frontline setting [7]. 

To improve the efficacy of ICIs, different immunotherapy combination approaches have been evaluated. Current treatment strategies include pairing ICIs with anti-angiogenic agents targeting the vascular endothelial growth factor (VEGF), or using dual immune checkpoint blockade. The combination of atezolizumab (an anti-PD-L1antibody) and bevacizumab (an anti-VEGF antibody) became a new standard of care in the first-line setting for advanced HCC based on an unprecedented overall (OS) benefit [8]. Dual checkpoint blockade with durvalumab (an anti-PD-L1 antibody) and tremelimumab (an anti-CTLA-4 antibody) also demonstrated significant survival benefit and has become an alternative first-line treatment, along with more recent data for ipilimumab (an anti-CTLA-4 antibody) and nivolumab (an anti-PD-1 antibody) [5,9]. These standard first-line therapies have transformed the therapeutic landscape of advanced HCC. In this review, we discuss the underlying rationale and mechanism of immunotherapeutic strategies, clinical efficacy data, and potential resistance mechanisms. We will also review the challenges in establishing predictive biomarkers for immunotherapy in advanced HCC and outline future directions in the field, with a focus on the novel immune checkpoints under investigation. 

## 2. Hepatocarcinogenesis and the Tumor Microenvironment in HCC

HCC is an inflammation-driven malignancy, involving both genetic and epigenetic changes. The tumorigenesis of HCC involves a multistep progression with the most common sequence beginning with liver injury and chronic inflammation. This triggers liver cell death and regeneration, eventually leading to fibrosis, cirrhosis, the development of dysplasia, and ultimately HCC [10]. Although more than 80% of HCC develops in the setting of cirrhosis, an exception to this pathway has been with hepatitis B virus (HBV), as virus integration may lead to chronic hepatitis and development of HCC in the absence of cirrhosis [11]. This may also occur in patients with nonalcoholic fatty liver disease (NAFLD), with approximately 20% of NAFLD-related HCC occurring without underlying cirrhosis [12].

A healthy liver is known to be immunologically tolerant. Approximately 80% of its blood supply originates from the portal vein, which transports high levels of pathogenic antigen loads via first-pass metabolism. To avoid hepatic autoimmunity, the liver exists in a delicate balance of immunosurveillance while maintaining self-tolerance. Various resident liver cells, such as Kupffer cells, hepatocytes, and liver sinusoidal epithelial cells, produce regulatory cytokines that contribute to this milieu [13]. 

Although HCC develops in a background of inflammation and is often marked by the presence of tumor infiltrating lymphocytes (TILs), its tumor microenvironment (TME), is largely immunosuppressive. The HCC TME comprises a unique landscape of immune cells, cytokines, and tumor-associated signaling pathways. HCC tumor cells may further potentiate the immunosuppressive conditions of the healthy liver through recruitment of regulatory T cells (Tregs), myeloid-derived suppressor cells (MDSCs), and tumor-associated macrophages (TAMs), contributing to HCC’s immune evasion [14]. Increased Tregs in patients with HCC have been associated with CD8+ T cell dysfunction and worsened survival [15]. Similarly, increased MDSCs suppress T cells through the production of suppressive cytokines, such as transforming growth factor-β (TGF-β) [16]. Additional immune changes, including dysfunctional dendritic cells, CD8+PD1+ T cells, neutrophils, and regulatory B cells producing metalloproteinases, further impact the HCC TME. In addition, the HCC TME is also characterized by an upregulation of well-known immune checkpoints, such as PD-1 and CTLA-4 [14]. Given this immunosuppressive milieu, immune checkpoint inhibitors, which increase T cell activation against tumor cells blocking inhibitory immune checkpoints, have become an effective approach to harnessing the immune response against HCC. 

In addition to a microenvironment that supports immune suppression, HCC is a highly vascular type of tumor, mediated by VEGF and other pro-angiogenic mediators, including basic fibroblast growth factor, tumor necrosis factor alpha (TNFα), transforming growth factor-β, platelet-derived growth factor (PDGF), and platelet growth factor (PGF) [17]. The targeted inhibition of the VEGF/vascular endothelial growth factor receptors (VEGFR) pathway has demonstrated clinical efficacy in advanced HCC [18,19,20,21,22]. Furthermore, VEGF inhibition may have immunomodulatory effects. Bevacizumab, an anti-VEGF monoclonal antibody, has been shown to prevent Treg accumulation and induce “vessel normalization”, improving the trafficking of T cells to the tumor site [23]. Sorafenib, a multi-targeted tyrosine kinase inhibitor which inhibits VEGFR and other pathways, has been shown to significantly downregulate Tregs and MDSCs [24]. The combination of VEGF pathway inhibitors with ICIs has therefore been a focus of numerous clinical studies. 

## 3. Evolution of Systemic Treatment in HCC

Prior to the adoption of immunotherapy for advanced HCC, multiple oral anti-angiogenic TKIs were investigated. These agents inhibit multiple tyrosine kinase enzymes and commonly inhibit VEGFRs and platelet-derived growth factor receptor β (PDGFR-β), but they differ in the degree of inhibition of these targets or their additional effects on other angiogenesis pathways [18,19,20,21]. 

Based on the phase III, Sorafenib Hepatocellular Carcinoma Assessment Randomized Protocol [25] trial, and the follow-up Asia–Pacific trial, sorafenib became the first targeted agent to show an OS benefit for patients with advanced HCC [18,26]. This agent exhibits both anti-tumor proliferation and anti-angiogenic effects, including the inhibition of VEGFR 1-3, RAF kinases (RAF-1 and B-RAF), and PDGFR-β. In the SHARP trial, patients receiving sorafenib demonstrated an improvement in OS compared with placebo, from 7.9 to 10.7 months (hazard ratio [HR] 0.69, 95% confidence interval [CI] 0.55–0.87), and an improvement in time to radiologic progression from 2.8 to 5.5 months (HR 0.58, 95% CI 0.45–0.74) [18]. 

Lenvatinib, another oral TKI targeting VEGFR 1-3 and fibroblast growth factor receptors 1-4, emerged as another frontline option for advanced HCC based on the phase III REFLECT trial. Lenvatinib demonstrated non-inferiority to sorafenib in terms of OS (13.6 months vs. 12.3 months, respectively; HR 0.92, 95% CI 0.79–1.06) but a superior objective response rate (ORR) of 24.1% vs. 9.2% and a progression-free survival (PFS) of 7.4 months vs. 3.7 months, HR 0.66, 95% CI 0.57–0.77, establishing lenvatinib as an alternative to sorafenib in the first-line setting [19]. Other studies have evaluated TKIs after progression on sorafenib, including regorafenib [20] and cabozantinib [21], as well as ramucirumab, an anti-VEGFR-2 monoclonal antibody [22]. 

## 4. Development of Immunotherapy in HCC

The ability of cancer to evade the normal immune response involves the adaptive upregulation of immune checkpoints. In tumors, the interaction between PD-1, an immune-inhibitory receptor on activated T cells, with its ligand PD-L1 on tumor cells, inhibits tumor cell apoptosis and promotes T cell exhaustion. This interaction creates a physiological “brake” on the immune response to cancer. 

Beyond tumor cells, preclinical and translational studies have shown that PD-L1 is also upregulated in various cell types within the HCC TME. For example, Kupffer cells in the tumor stroma exhibit an increased PD-L1 expression, which correlates with poorer prognosis. Blocking the PD-L1/PD-1 interaction between Kupffer cells and effector T cells restores effector T cell function [27]. Similarly, a proportion of monocytes and macrophages in the peritumoral stroma express PD-L1, with higher density associated with advanced disease and worsened survival outcomes. These PD-L1 expressing monocytes suppress tumor-specific T cell immunity, contributing to tumor growth in vivo, which may be reversed with PD-L1 inhibition [28]. 

The development and clinical use of PD-1 or PD-L1 inhibitors, such as nivolumab and pembrolizumab, permit activated T cells to overcome these immune regulatory mechanisms and mount anti-tumor responses [29]. The CheckMate-040 study, a multicohort phase I-II trial of nivolumab in advanced HCC in both sorafenib naïve and experienced patients, was one of the key initial clinical studies to demonstrate a potential durable survival benefit in a proportion of patients, with responses seen regardless of HCC etiology or PD-L1 expression [30]. In this trial, nivolumab demonstrated an acceptable safety profile with a response rate of up to 20% in the dose expansion cohort.

Numerous randomized phase III trials investigating single-agent anti-PD-1 antibodies subsequently ensued. The CheckMate-459 trial, comparing nivolumab to sorafenib as first-line treatment in advanced HCC, was unable to show a significant improvement in OS of nivolumab (16.4 vs. 14.7 months, HR 0.85, 95% CI 0.72–1.02), despite a higher response rate (15% vs. 7%) and a favorable safety profile (Table 1). Possible reasons for the lack of OS benefit include the higher percentage of patients in the sorafenib arm receiving subsequent immunotherapy as well as time-varying hazard ratios with delayed separation of Kaplan–Meier curves [7]. Similarly, the RATIONALE-301 study, which compared the anti-PD-1 antibody tislelizumab with sorafenib in the frontline setting for advanced HCC in a predominantly Asian population, showed non-inferiority but not superiority in OS (15.9 vs. 14.1 months, HR 0.85, 95% CI 0.71–1.02) [6]. Additionally, the KEYNOTE-240 trial, which randomized patients who had progressed on prior sorafenib to the anti-PD-1 antibody pembrolizumab or best supportive care (BSC), did not meet the prespecified statistical significance criteria for OS or PFS superiority [31]. In contrast, the KEYNOTE-394 trial in an Asian population of patients with advanced HCC who had progressed on or were intolerant to sorafenib or chemotherapy incorporating oxaliplatin (an alkylating agent) showed a statistically significant, though modest, improvement in OS from 13.0 to 14.6 months with pembrolizumab (HR 0.79, 95% CI: 0.63–0.99), as well as improvement in PFS and ORR [32]. Thus, single-agent anti-PD-1 inhibitors have achieved a variable, though mostly disappointing, degree of success when compared with anti-angiogenic TKIs.

## 5. Mechanisms of Resistance to Immunotherapy

The limited effectiveness of single-agent ICIs may be related to various known resistance mechanisms, which may be classified as primary resistance (no response to immunotherapy) or acquired resistance (initial response followed by progression or relapse). 

### 5.1. Primary Resistance

Primary resistance mechanisms are impacted by factors that are either tumor cell-intrinsic, involving changes within tumor cells that prevent immune cell infiltration or function, or tumor cell-extrinsic, involving the TME and host-related factors leading to immunosuppression [33]. 

### 5.2. Tumor-Intrinsic Mechanisms

Tumor cell-intrinsic mechanisms, which may exist from the beginning or evolve later and serve as an adaptive resistance mechanism, include low tumor immunogenicity, dysfunction of neo-antigen presentation by tumor cells, and alterations in key signaling pathways [34]. 

The ability of the immune system to recognize and distinguish neo-antigens (tumor-specific antigens) from normal tissue is critical in the immune response. This tumor immunogenicity may be impacted by tumor somatic mutational burden (TMB), which may correlate with neo-antigen burden and frequency. However, HCC is not typically characterized by high TMB. In a cohort of 755 patients with HCC who underwent genomic profiling, Ang et al. reported a frequency of only 0.8% of patients with high TMB [35]. Moreover, discrepant results of tumor response to immunotherapy based on TMB are likely attributed to the lack of consistent correlation between TMB and neo-antigen load [36]. Although tumor neo-antigen burden (TNB) may better correlate with immunotherapy response outcomes, assays and cutoffs for TNB measurement are not definitively established and therefore not yet adopted into standard use [37]. 

Furthermore, dysfunction of neo-antigen presentation and evasion of T cell recognition may be facilitated by the decreased expression of neo-antigens via mechanisms such as hypermethylation and the subsequent silencing of genes [38]. Mutations in genes such as beta-2 microglobulin (β2m) which are involved in major histocompatibility complex (MHC) presentation may also negatively inhibit antigen-presenting mechanisms and confer resistance to immunotherapy [39]. 

Finally, aberrations in signaling pathways have been implicated in intrinsic resistance mechanisms. Defects in Janus tyrosine kinases (JAK) and signal transducer and activator of transcription (STAT) pathways affect interferon-alpha (IFN-α)-induced priming of T cells and the subsequent activation of effector T cells [40]. Alterations in the WNT/β-catenin pathway may impair the downstream recruitment and infiltration of T cells and the suppression of natural killer (NK) cell activity [41,42,43]. In addition, impairments in phosphatase and tensin homologue (PTEN) and signal transducer and activator of transcription factor 3 (STAT 3) pathways have been associated with an increase in specific cytokines, which promote an immunosuppressive TME [44,45].

### 5.3. Tumor-Extrinsic Mechanisms

Tumor cell-extrinsic mechanisms primarily relate to components of the tumor microenvironment (TME) which create an immunosuppressive state. These include the activation of cell populations (Tregs, MDSCs, TAMs), T cell exhaustion, and the activation of alternative immune checkpoint pathways [46]. 

Tregs, which normally function in autoimmunity, homoeostasis, and self-tolerance, demonstrate immunosuppressive activity in the context of cancer and promote tumor immune invasion [47]. This is in part due to effect of anti-inflammatory signaling molecules which allow for Treg maturation, such as IL-10, on downregulating MHCII on APCs, and decreasing the production of proinflammatory cytokines such as IL-2 [48]. Tregs also overtake the utilization of IL-2 in the TME, which in turn negatively impacts the proliferation and activation of T cells and natural killer (NK) cells. The combination of these mechanisms effectively blunts the antitumor immune response [49,50]. MDSCs inhibit T and NK cell effector function by leading to high expression of enzymes such as ARG1, which consumes L-arginine, a molecule essential for T cell proliferation, and ultimately causing downstream apoptosis of tumor infiltrating lymphocytes (TILs) [16,51]. MDSCs also lead to the increased production of indoleamine 2,3-dioxygenase (IDO), which depletes L-tryptophan from T cells, promoting the formation of T cells exhibiting Treg behavior [52]. Furthermore, TAMs express cytokines and chemokines that suppress anti-tumor immunity and promote progression [53]. 

T cell exhaustion is due to chronic antigen stimulation leading to T cell dysfunction [54]. The activation of alternative immune checkpoints may also contribute to T cell exhaustion. These include T cell immunoreceptor with immunoglobulin and ITIM domain (TIGIT), a receptor expressed on CD4, C8, and NK cells, which may impair T cell receptor activation, T cell immunoglobulin, and mucin-domain containing-3 (TIM-3), a coinhibitory protein expressed by dysfunctional CD8+ TILs, and lymphocyte activation gene 3 (LAG-3), an inhibitory receptor on activated T lymphocytes which interferes with bindings of MHCII to T cell receptor and prevents cytokine release [25,55,56].

### 5.4. Acquired Resistance

The mechanisms of the acquired ICI resistance, specifically within HCC, remain unclear; however, certain pathways have been observed in other tumor types. The loss of encoding sequences for tumor neo-antigens in tumor cells has been demonstrated to lead to the production of less immunogenic tumor cell clones evading T cell killing [57]. An acquired loss of β2m resulted in the decreased expression of HLA and impaired antigen presentation in lung cancer [58]. In melanoma, acquired resistance has also been attributed to JAK 1/2-inactivating mutations [59]. Further research in clinical studies comprising larger cohorts of patients is needed to better elucidate the underlying acquired resistance mechanisms in HCC.

## 6. Combination Strategies with Immunotherapy to Overcome Resistance

Combination immunotherapy treatment strategies have shown superior clinical efficacy, with ICIs combined with VEGF arising as a key therapeutic approach. VEGF overexpression in HCC promotes angiogenesis and contributes to immunosuppressive TME. The inhibition of the VEGF/VEGFR pathway facilitates the transient restoration of proper, mature blood vessels to the tumor, termed “vascular normalization”, increasing drug delivery and cytotoxic T cell infiltration to the tumor [23]. Additionally, VEGF produced by tumor and other cells, contribute to an immunosuppressive TME through several mechanisms that affect immune cell function. For example, VEGF-A has been shown to increase PD-1 expression on activated CD8+ T cells, which may be effectively blocked by anti-VEGF-A antibodies in vitro [60]. Motz et al. demonstrated that VEGF induces Fas Ligand (FasL) expression on tumor endothelial cells, leading to the preferential elimination of CD8+ T cells. Blocking VEGF led to a significant increase in tumor-infiltrating CD8+ T cells, with no change in the numbers of FOXP3+ T regs [61]. Beyond modulating T cell trafficking, VEGF produced by cancer cells may also impact early stages of dendritic cell maturation from CD34+ precursors, affecting their morphology and impairing antigen presentation [62]. Moreover, blocking VEGF has been shown to reduce the infiltration of suppressive MDSCs and Tregs while increasing the fraction of mature dendritic cells [63]. Thus, the benefit of VEGF inhibition extends beyond effects on angiogenesis, and combining VEGF/VEGFR blockade with PD-1/PD-L1 inhibition may potentiate the efficacy of ICI therapy.

The pivotal IMBrave-150 trial established a new standard of care with the combination of atezolizumab and bevacizumab for advanced HCC in the first-line setting, demonstrating OS superiority over sorafenib (19.2 vs. 13.2 months; HR 0.66, 95% CI 0.52–0.85), along with a significant improvement in PFS (6.9 vs. 4.3 months; HR 0.65, 95% CI 0.53–0.81). This combination also showed higher ORR of 30%, compared with 5% with sorafenib. The toxicity profile was favorable, with grade 3 and 4 treatment-related adverse events (TRAEs) in 43% of patients receiving the combination, compared with 46% with sorafenib, with improvements in the deterioration of quality of life (QOL). Aside from potential immune-related AEs, notable side effects of the combination were primarily related to bevacizumab, including proteinuria and hypertension [64] (Table 1). 

Given the established efficacy of TKIs as monotherapy, combinations of anti-PD-1/PD-L1 therapy with anti-angiogenic TKIs targeting the VEGF/VEGFR pathway were initially anticipated to carry similar or improved clinical efficacy, particularly in light of the success of IMbrave-150. TKIs have also shown immunomodulatory activity in preclinical studies. For example, lenvatinib has demonstrated immunomodulatory activity via CD8+ T cells populations and improved antitumor activity when combined with anti-PD-1 therapy [65], and cabozantinib induces tumor cell changes increasing sensitivity to T cell-mediated lysis [66]. Nonetheless, ICIs plus antiangiogenic TKIs have exhibited limited success in clinical studies. 

In the COSMIC 312 study, cabozantinib plus atezolizumab did not demonstrate superiority in OS compared with sorafenib as first-line therapy in patients with advanced HCC (OS 16.5 vs. 15.5 months, respectively; HR 0.98, 96% CI 0.78–1.24), but it improved PFS (6.9 vs. 4.3 months; HR 0.74, 95% CI: 0.56–0.97) [67,68] (Table 1). Similarly the LEAP-002 study, which compared lenvatinib plus pembrolizumab versus Lenvatinib, was unable to reach the promising efficacy initially demonstrated in an earlier phase study [69]. This study did not demonstrate superiority in PFS (8.2 vs. 8.0 months; HR 0.87, 95% CI:0.73–1.02) or OS (21.2 vs. 19.0 months; HR 0.84, 95% CI 0.71–1.00) with the combination (Table 1). On the other hand, the CARES-310 trial, investigating camrelizumab (an anti-PD-1 antibody) plus rivoceranib (a VEGFR2-targeted TKI, also known as apatinib), demonstrated improved OS (22.1 vs. 15.2 months; HR 0.62, 95% CI 0.49–0.80) and PFS (5.6 vs. 3.7 months; HR 0.52, 95% CI: 0.41–0.65) (Table 1). However, 83% of patients in this study were notably enrolled from Asia; thus, this combination has had limited adoption worldwide [70]. Reasons for the failures of COSMIC-312 and LEAP-002 to meet their primary endpoint of OS remain unclear. However, possible explanations include receipt of efficacious second-line therapeutic strategies in the control arm, including immunotherapy [71]. This may account for the superior PFS observed with atezolizumab/cabozantinib over sorafenib in COSMIC-312, but ultimately no OS benefit. Furthermore, the underlying etiology has been postulated to explain differences in the survival outcomes. Exploratory subgroup analyses of COSMIC-312 showed that patients with hepatitis B derived clearer PFS and OS benefit than other etiology subgroups [72]. These findings may similarly raise the question in CARES-310 of whether the OS superiority of camrelizumab/rivoceranib in the overall study population may have been driven by the benefit derived in the predominantly Asian population with Hepatitis B etiology. 

Beyond VEGF inhibition, dual checkpoint blockade incorporating CTLA-4 inhibition with PD-1 or PD-L1 antibodies has also significantly advanced the treatment landscape of advanced HCC. CTLA-4 receptors are constitutively expressed on Tregs, and expression on effector CD4+ and CD8+ T cells is induced following activation [73]. Once expressed, CTLA-4 outcompetes the costimulatory receptor CD28 for binding to CD80 and CD86 on antigen-presenting cells (APCs) [74]. In the TME, CTLA-4 is expressed on infiltrating Tregs, promoting immune escape [73]. Dual blockade of PD-1 and CTLA-4 in preclinical models led to effector T cell upregulation and inhibitor T cell downregulation [75]. Additionally, CTLA-4 inhibition has been shown to increase the effector T cell to MDSC ratios within tumors, potentially overcoming primary resistance to anti-PD-1 monotherapy, associated with the presence of MDSCs and Tregs [76]. CTLA-4 inhibition may also play a role in durable immunotherapy responses, with the proliferation of transition memory T cells [77]. 

Another practice changing study, the HIMALAYA trial demonstrated that the combination of durvalumab and a single priming dose of tremelimumab improved OS compared with sorafenib (16.4 vs. 13.8 months; HR 0.78, 96.02% CI 0.65–0.93). In addition, this study also established the non-inferiority of durvalumab monotherapy to sorafenib (HR 0.86, 95.67% CI: 0.73–1.03). No PFS benefit was observed with the ICI regimens, which may be related to the delayed benefit of ICI, given the later divergence of the survival curves. Patients receiving durvalumab/tremelimumab exhibited an ORR of 20.1%, compared with 5.1% in those receiving sorafenib, and a longer duration of response (DOR) at nearly 2 years [5]. Updated results at 4 years have shown a 25.2% survival rate with the combination [78]. In terms of AEs, about 20% of patients required high-dose steroids for immune-related AEs [5]. The combination regimen has become another frontline option for advanced HCC, and an alternative for patients with high bleeding risk or recent cardiovascular events.

The combination of nivolumab plus ipilimumab was also evaluated in the multicohort CheckMate-040 trial. In patients who had received prior sorafenib, this combination demonstrated an ORR of 32%, a median DOR of 17.5 months (95% CI: 5–47+), and OS of 22.8 months (95% CI: 9.4-NE) after a minimum follow-up of 44 months [79]. These findings led to the FDA approval of this regimen for advanced HCC after progression or intolerance to sorafenib. Notably, however, 53% of patients experienced grade 3–4 TRAEs [80]. Salvage ipilimumab and nivolumab after prior anti-PD(L)-1 therapy has also been investigated in advanced HCC, suggesting that prior anti-PD-(L)-1 therapy may not preclude response to dual ICI blockade in later lines of therapy. Anti-CTLA-4 therapy may help overcome prior resistance to anti-PD-(L)-1 monotherapy [81]. 

The phase III CheckMate-9DW study subsequently compared nivolumab/ipilimumab with sorafenib or lenvatinib in the first-line setting for patients with advanced HCC. The trial noted a significant improvement in OS from 20.6 months with TKI to 23.7 months with the ICI combination (HR 0.79, 95% CI 0.65–0.96). Nivolumab/ipilimumab also demonstrated one of the highest response rates (36%) in a phase III trial in advanced HCC. As observed in CheckMate-040, there was a relatively high rate of grade 3–4 TRAEs occurring in 41% of patients in the combination arm, with 18% having TRAEs leading to discontinuation, and 4% with treatment-related deaths [9]. Given these data, nivolumab and ipilimumab may be considered as an additional first-line standard of care option for advanced HCC. 

## 7. Predictive Biomarkers in HCC: Challenges and Progress

Despite the evolving treatment landscape in advanced HCC, there remains a critical need to identify biomarkers which predict response to certain therapies in order to tailor therapy selection for patients, enhance clinical outcomes, and prevent patients from experiencing unnecessary toxicity. The pathogenesis of HCC is multifaceted, and genetic heterogeneity and different underlying disease etiologies have posed significant challenges in the identification of effective biomarkers. 

Although PD-L1 expression has been utilized as a predictive biomarker for treatment effectiveness of anti-PD-(L)-1 therapy in other cancers, such as lung cancer [82] and gastroesophageal cancer [83], its prognostic and predictive value in HCC has been less conclusive. Some clinical trials such as CheckMate-459, the phase III study of nivolumab versus sorafenib as first-line treatment of advanced HCC, showed that positive tumor cell PD-L1 expression correlated with higher treatment response in patients receiving nivolumab [7]. In this study, patients with PD-L1 expression ≥ 1% demonstrated ORR of 28%, compared with ORR of 12% in patients with PD-L1 < 1%. However, other studies such as CheckMate-040, which preceded CheckMate-459 and evaluated both nivolumab monotherapy and nivolumab plus ipilimumab combination therapy in different study cohorts, did not show a clear predictive effect of tumor cell PD-L1 expression on response. In its dose expansion phase of the nivolumab monotherapy cohort, the ORR was 26% in patients with PD-L1 expression ≥ 1% versus 19% in patients with PD-L1 < 1%, suggesting response regardless of PD-L1 status [3,80]. Thus, there have been conflicting, inconsistent results regarding PD-L1, and ultimately, it has not been a reliable, predictive biomarker in HCC.

Other molecular biomarkers predictive of immunotherapy response in solid tumors, such as tumor mutation burden (TMB)-high, microsatellite instability (MSI)-high, and POLE and POLD mutations, also play a limited role in HCC due to their overall low prevalence. In a cohort of patients with HCC who underwent comprehensive genomic profiling, 95% of patient tumors had a low TMB of <10 mutations/Mb, with a median TMB of 4 mutations/Mb [35]. MSI-H was noted in only 1 of 542 patients, and POLE and POLD alterations were noted in 4% of patients [35]. Moreover, in an analysis by Zhu et al. of IMbrave-150 data, when correlating clinical outcomes with tertiles of TMB levels, there was no clear association between TMB and response rate or survival benefits in patients receiving atezolizumab/bevacizumab [84].

Recent studies suggest that the characteristics of the TME may serve as a potential biomarker of immunotherapy efficacy. A post hoc analysis of tumor samples from patients with HCC enrolled in the GO30140 phase 1b trial and the IMbrave-150 trial showed that molecular correlates associated with improved clinical response to atezolizumab/bevacizumab included pre-existing immunity (high expression of CD274, T-effector signature and intratumoral CD8+ T cell density), the high expression of VEGFR-2, T regulatory cells, and myeloid inflammatory signatures. The reduced clinical benefit of this combination was associated with high regulatory T-reg to effector T cell ratio and expression of genes, such as glypican 3 (GPC3) and alpha-fetoprotein (AFP) [84]. 

Beyond molecular and immune biomarkers, there has also been a question of whether the etiology of HCC impacts the efficacy of immunotherapy. Although subgroup analyses of survival in IMbrave-150 suggested potentially greater benefit with atezolizumab/bevacizumab in patients with viral hepatitis compared with patients without underlying viral etiology, the real-world analysis of the combination did not suggest a clear association on survival [8,85]. A post hoc analysis of IMbrave-150 also did not demonstrate differences in OS, PFS, or ORR between different etiologies, such as HBV, HCV, and alcohol-associated and fatty liver diseases [86]. In the HIMALAYA trial, sub-group analyses suggested greater OS benefit in patients with HBV compared with HCV; however, patients with non-viral etiology still demonstrated survival gain [5]. A meta-analysis by Ding et al. further explored the impact of viral etiology on the efficacy of immunotherapy in HCC and found no significant difference in response rates between patients with underlying viral etiology versus those without, as well as similar response rates between patients with HBV and HCV [87].

Through biospecimen collection and correlative analyses, the NCI-CLARITY (National Cancer Institute Cancers of the Liver: Accelerating Research of Immunotherapy by a Transdisciplinary Network), an ongoing prospective study, seeks to understand the mechanisms of response and resistance to immunotherapy in patients with hepatobiliary cancers undergoing treatment with first-line immunotherapy [88]. These efforts underscore the continued need to develop novel biomarkers to enhance predictive accuracy in treatment selection and response to ultimately improve patient outcomes in HCC. Nonetheless, until there is clear validation of biomarkers in larger cohorts of patients in future research, there currently remains no recommendations for specific markers to be adopted and integrated into widespread clinical practice. 

## 8. Immunotherapy in HCC: Future Directions and Challenges

Novel immune checkpoints in HCC are currently under investigation, with several promising targets. Building upon the success of IMbrave-150, the IMbrave-152 is a randomized phase III trial evaluating the combination of atezolizumab and bevacizumab, with or without tiragolumab, a monoclonal antibody to TIGIT in the first-line setting, to determine whether the present standard of care may be further enhanced [89]. TIGIT downregulates T cell and natural killer (NK) cell function by interacting with CD-155 on antigen-presenting cells and tumor cells [90]. Its inhibition may enhance CD8+ T cell and NK cell responses while reducing T reg suppression [91]. In preclinical studies, although TIGIT inhibition alone had minimal effects on tumor growth, the dual blockade of PD-1 and TIGIT acted synergistically to overcome resistance mechanisms to anti-PD-(L)1 therapy [91]. In the phase IB/II MORPHEUS-liver study, triplet therapy with atezolizumab, bevacizumab, and tiragolumab compared with standard atezolizumab/bevacizumab demonstrated a significantly higher ORR (42.5% vs. 11.1%) and a higher PFS of 11.1 vs. 4.2 months (HR 0.42, 95% CI 0.22–0.82) [92] (Figure 1). 

This figure shows ongoing immunotherapy clinical trials in HCC. Treatments are grouped into three categories as follows: immunotherapy alone, immunotherapy combined with tyrosine multikinase inhibitors, and immunotherapy combined with anti-VEGF antibodies. The figure also indicates the line of therapy and the phase of each trial. Abbreviations: CAR-T, chimeric antigen receptor (CAR) T cell therapy; CTLA-4, cytotoxic T-lymphocyte-associated antigen 4; IL-27, interleukin 27; LAG-3, lymphocyte activation gene-3; PD-1, programmed cell death protein 1; PD-L1, programmed cell death ligand 1; TIGIT, T cell immunoreceptor with immunoglobulin and tyrosine-based inhibitory motif (ITIM) domain; TIM-3, T cell immunoglobulin and mucin-domain-containing-3; and VEGF, vascular endothelial growth factor

Another emerging immune checkpoint target is LAG-3, an inhibitory receptor that may regulate and inhibit T cell activation [93]. LAG-3 expression is upregulated in tumor infiltrating CD8+ T cells in a variety of tumor types including HCC [94,95]. In one study of tumor samples from resected HCC patients, the expression of LAG-3 on TILs was found in 65% of samples along with expression of PD-L1 in 83% [95]. Multiple trials combining various LAG-3 inhibitors with anti-PD-1 therapy in advanced HCC are ongoing [96,97]. In addition, studies are also evaluating the potential of LAG-3 as a predictive biomarker for treatment response [98,99]

TIM-3, another immune checkpoint receptor under investigation, plays a key role in T cell exhaustion [100]. The expression of TIM-3, as well as PD-1, LAG3, and CTLA-4, are higher on T cells in HCC tumor tissue compared with normal liver tissue [101]. In patients with HBV-associated HCC, TIM-3 and PD-1 were upregulated on CD4+ and CD8+ TILs, significantly decreasing the secretion of interferon gamma (IFN-γ) and tumor necrosis factor alpha (TNF-α). The in vitro inhibition of TIM-3 and PD-1 enhanced TIL proliferation and IFN-γ and TNF-α secretion [102]. A planned phase IB/II trial is evaluating the safety and preliminary efficacy of a TIM-3 inhibitor with durvalumab in patients with advanced HCC [103]. 

Beyond checkpoint inhibitors, chimeric antigen receptor T cell (CAR-T) therapy has revolutionized the treatment of hematological malignancies and is being investigated as a treatment modality for solid tumor malignancies, including HCC. In this cellular therapy approach, T cells isolated from a patient or allogeneic donor are genetically modified to express chimeric antigen receptors (CARs), which allow for the T cell-targeted cytotoxicity of malignant cells. The CAR-T cells are then expanded and infused into the patient [104]. Several CAR-T targets in HCC are under investigation, including GPC-3, AFP, CD133, and c-Met [105]. GPC-3 is upregulated in a variety of tumors including HCC with restricted expression in normal tissues [106]. In in vitro and xenograft models, GPC-3-targeted CAR demonstrated cytotoxic activity against GPC3-positive human HCC cells [107]. There are multiple ongoing trials of GPC-3-directed CAR-T in patients with advanced HCC [108,109,110]. 

Despite substantial progress, advances in immunotherapy in HCC have lagged behind other malignancies, such as melanoma and non-small cell lung cancer (NSCLC). One difference is the earlier integration of immunotherapy in melanoma and NSCLC, including the neoadjuvant and adjuvant settings. The preoperative and postoperative combination immunotherapy strategies have not yet been incorporated into standard practice in HCC. The randomized IMbrave-50 trial evaluating adjuvant atezolizumab/bevacizumab versus observation following resection or ablation in patients with “high-risk” HCC showed no improvement in relapse free survival (RFS) (median RFS 33.2 months with the combination vs. 36.0 months with active surveillance, HR 0.90 [95% CI 0.72–1.12]) after extended follow-up, with mature OS data still pending [111]. However, more recent research has suggested that combined modality therapy for unresectable HCC without extrahepatic spread may improve clinical outcomes. In the EMERALD-1 study, patients with unresectable, liver-confined HCC amenable to transarterial chemoembolization (TACE) who received TACE plus durvalumab/bevacizumab showed superior PFS compared with those treated with TACE alone (median PFS 15.0 vs. 8.2 months, HR 0.77 (95% CI 0.61–0.98, *p* = 0.032) [112]. Similarly, the LEAP-002 study investigating pembrolizumab/lenvatinib plus TACE versus TACE alone showed a significant improvement in PFS for the combination (median PFS 14.6 vs. 10.0 months, HR 0.66 (95% CI 0.51–0.84, *p* = 0.0002). However, further data are needed to clarify whether combined modality therapy in these studies ultimately improves the overall survival of patients.

Finally, beyond clinical efficacy, patient selection criteria in clinical trials for advanced HCC have raised concerns regarding the applicability of newer treatment options for all patients in a real-world setting. This is largely due to the inclusion of patients with preserved liver function alone (i.e., Child-Pugh A) in most clinical studies, due to concerns regarding toxicity and tolerability with decompensated liver disease, where liver dysfunction may be a competing factor in a patient’s morbidity and mortality. Data supporting the utilization of ICI monotherapy in Child-Pugh B patients has been mainly derived from retrospective studies, and robust treatment guidelines for this population are lacking. While these studies have generally shown overall tolerability to treatment, outcomes are often less favorable [113]. Ongoing studies are being conducted to evaluate the safety and efficacy of combination immunotherapy strategies such as atezolizumab/bevacizumab in patients with advanced HCC and Child-Pugh B cirrhosis [114]. Future trials must balance patient selection with inclusive access to treatment, particularly in patients with impaired or borderline liver function who may still derive clinical benefit from novel therapies.

## 9. Conclusions

The incorporation of immunotherapy in advanced HCC has evolved significantly over the past decade, fundamentally changing the treatment paradigm. ICI monotherapy with PD-1 inhibitors such as nivolumab or pembrolizumab initially demonstrated modest response rates, and are susceptible to a variety of resistance mechanisms, including altered antigen presentation or signaling pathways, the expression of immunosuppressive cell populations, and the activation of alternative immune checkpoint pathways. The combination of anti-PD-1 plus anti-VEGF inhibition with atezolizumab/bevacizumab harnesses the immunomodulatory effect of anti-angiogenic therapy to enhance the efficacy of ICI therapy. However, studies with ICIs and anti-angiogenic TKIs have had comparatively mixed success. Camrelizumab/rivoceranib remains the only ICI plus TKI combination to demonstrate superiority in survival over sorafenib; however, further validation of its efficacy is needed in a more diverse, global patient population. Dual immune checkpoint blockade with the incorporation of anti-CTLA-4 inhibition may help overcome certain resistance mechanisms to PD-(L)1 inhibitors. Both durvlumab/tremelimumab and nivolumab/ipilimumab have demonstrated the potential for durable responses and long-term survival to immunotherapy. Persistent challenges exist in the development and incorporation of predictive biomarkers for immunotherapy into clinical practice in advanced HCC; however, the exploration of immune inflammatory signatures may hold promise. Ongoing and future research efforts have focused on investigating novel immune checkpoints, such as TIGIT, LAG-3, and TIM-3, as well as newer therapies such as CAR-T targeting GPC-3, AFP, and other proteins expressed on HCC tumors, potentially providing greater advances and the expansion of the therapeutic armamentarium in HCC.

## Figures and Tables

**Figure 1 ijms-26-01936-f001:**
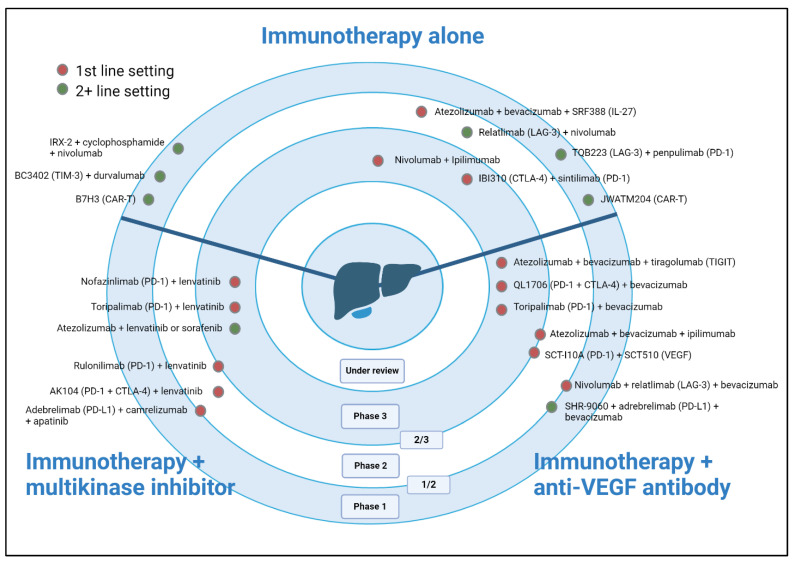
Selected ongoing clinical trials in patients with advanced HCC.

**Table 1 ijms-26-01936-t001:** Selected Frontline Immunotherapy Trials in Advanced HCC.

	IMbrave150		HIMALAYA			CheckMate9DW		CARES-310		LEAP-002		COSMIC-312			CheckMate459	
	(NCT03434379)		(NCT03298451)			(NCT04039607)		(NCT03764293)		(NCT03713593)		(NCT03755791)			(NCT02576509)	
**Trial Type**	Phase III RCT		Phase III RCT			Phase III RCT		Phase III RCT		Phase III RCT		Phase III RCT			Phase III RCT	
**Number of Patients**	501		1171			668		543		794		837			743	
**Eligibility**	CP-A, PS 0/1		CP-A, PS 0/1			CP-A, PS 0/1		CP-A, PS 0/1		CP-A, PS 0/1		CP-A, PS 0/1			CP-A, PS 0/1	
**Treatment**	Atezo/Bev	Sor	Durva/Treme	Durva	Sor	Nivo/Ipi	Sor	Cam/Rivo	Sor	Pem/ Len	Len	Atezo/Cabo	Sor	Cabo	Nivo	Sor
**Randomization**	1:1		1:1:1			1:1		1:1		1:1		2:1:1			1:1	
**OS (mos)**	19.2	13.4	16.4	16.6	13.8	23.7	20.6	22.1	15.2	21.2	19.0	16.5	15.5	14.5	16.4	14.7
**HR** **(95% CI)**	0.66 (0.52–0.85)		Durva/Treme vs. Sor: 0.78 (0.65–0.93) Durva vs. Sor: 0.86 (0.73–1.03) *			0.79 (0.65–0.96)		0.62 (0.49–0.80)		0.84 (0.71–1.00)		Atezo/Cabo vs. Sor: 0.98 (0.78–1.24) Cabo vs. Sor: 1.11 (0.85–1.43)			0.85 (0.72–1.02)	
**PFS (mos)**	6.9	4.3	3.8	3.6	4.1	9.1	9.2	5.6	3.7	8.2	8.1	6.9	4.3	5.8	3.7	3.8
**HR** **(95% CI)**	0.65 (0.53–0.81)		Durva/Treme vs. Sor: 0.90 (0.77–1.05) Durva vs. Sor: 1.02 (0.88–1.19)			0.87 (0.72–1.06)		0.52 (0.41–0.65)		0.83 (0.73–1.02)		Atezo/Cabo vs. Sor: 0.74 (0.56–0.97) Cabo vs. Sor: 0.78 (0.56–1.09)			0.93 (0.79–1.10)	
**ORR (%)**	30	11	20.1	17	5.1	36	13	25	6	26.1	17.5	13.0	4.6	7.4	15	7
**DCR (%)**	74	55	60.1	54.8	60.7	68	75	78	54	81.3	78.4	78	65	84	55	58
**DOR (mos)**	18.1	14.9	22.3	16.8	18.4	30.4	12.9	14.8	9.2	16.6	10.4	10.6	8.8	15.1	23.3	23.4

* non-inferiority margin met. Abbreviations: Atezo, Atezoliumab; Bev, Bevacizumab; Sor, Sorafenib; Durva, Durvalumab; Treme, Tremelimumab; Nivo, Nivolumab; Ipi, Ipilimumab; Cam, Camrelizumab; Rivo, Rivoceranib; Pem, Pembrolizumab; Len, Lenvatinib; Cabo, Cabozantinib; OS, overall survival; PFS, progression free survival; HR, hazard ratio; ORR, objective response rate; DCR, disease control rate; DOR, duration of response; CI, confidence interval; CP-A, Child-Pugh A score; PS, ECOG performance status; RCT, randomized controlled trial.

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
