# Peer review of "Advances in Immunotherapy in Hepatocellular Carcinoma"

_ijms, 2025, doi:10.3390/ijms26051936_

Round 1
Reviewer 1 Report
Comments and Suggestions for Authors
In the submitted manuscript, Bloom et al. reviewed the advances in immunotherapy for hepatocellular carcinoma (HCC). The authors give a clear and comprehensive summary of how immune checkpoint inhibitors (ICIs) and their combinations with other therapies could be used to treat HCC. Importantly, the authors highlighted recent clinical studies, which provides important insights for future therapies. The authors also acknowledge the limitations of single-agent ICI therapies and discuss potential resistance mechanisms. Overall, I think the authors provides a thorough and concise review of the existing literature in the field. The manuscript is well-structured and written. I believe the scope of the paper will be of interest to a broad readership of the journal.
To further improve the manuscript for final publication, I suggest the following minor revisions:
1. Could the authors further expand the discussions by including preclinical studies and translational research? This may provide a more comprehensive overview.
Author Response
Comment 1: Could the authors further expand the discussions by including preclinical studies and translational research? This may provide a more comprehensive overview.
Response to Comment 1: We appreciate the reviewer’s suggestion to incorporate preclinical and translation research for a more comprehensive discussion. We have added to the manuscript relevant preclinical studies exploring PD-L1 expression within the tumor microenvironment [Lines 141-149]:
“Beyond tumor cells, preclinical and translational studies have shown that PD-L1 is also upregulated in various cell types within the HCC TME. For example, Kupffer cells in the tumor stroma exhibit increased PD-L1 expression, which correlates with poorer prognosis. Blocking the PD-L1/PD-1 interaction between Kupffer cells and effector T-cells restores effector T-cell function[27]. Similarly, a proportion of monocytes and macrophages in the peritumoral stroma express PD-L1, with higher density associated with advanced disease and worsened survival outcomes. These PD-L1 expressing monocytes suppress tumor-specific T-cell immunity, contributing to tumor growth in-vivo, which may be reversed with PD-L1 inhibition[28]. “
Additionally, we have provided a more in-depth discussion of the VEGF pathway, highlighting its role in modulating T-cell function, dendritic cell maturation, and the infiltration of MDSCs [Lines 259-278]:
“Combination immunotherapy treatment strategies have shown superior clinical efficacy, with ICIs combined with VEGF arising as a key therapeutic approach. VEGF overexpression in HCC promotes angiogenesis and contributes to an immunosuppressive TME. Inhibition of the VEGF/VEGFR pathway facilitates transient restoration of proper, mature blood vessels to the tumor, termed “vascular normalization”, increasing drug delivery and cytotoxic T-cell infiltration to the tumor[23]. Additionally, VEGF produced by tumor and other cells, contribute to an immunosuppressive TME through several mechanisms that affect immune cell function. For example, VEGF-A has been shown to increase PD-1 expression on activated CD8+ T-cells, which may be effectively blocked by anti-VEGF-A antibodies in vitro[61]. Motz et al demonstrated that VEGF induces Fas Ligand (FasL) expression on tumor endothelial cells, leading to preferential elimination of CD8+ T-cells. Blocking VEGF led to a significant increase in tumor-infiltrating CD8+ T-cells, with no change in the numbers of FOXP3+ T regs[62]. Beyond modulating T-cell trafficking, VEGF produced by cancer cells may also impact early stages of dendritic cell maturation from CD34+ precursors, affecting their morphology and impairing antigen presentation[63]. Moreover, blocking VEGF has been shown to reduce the infiltration of suppressive MDSCs and Tregs, while increasing the fraction of mature dendritic cells[64]. Thus, the benefit of VEGF inhibition extends beyond effects on angiogenesis, and combining VEGF/VEGFR blockade with PD-1/PD-L1 inhibition may potentiate the efficacy of ICI therapy.”
Further discussion of preclinical supporting supporting CTLA-4 inhibition with dual immune checkpoint blockade is noted in Lines 326-337:
“Beyond VEGF inhibition, dual checkpoint blockade incorporating CTLA-4 inhibition with PD-1 or PD-L1 antibodies has also significantly advanced the treatment landscape of advanced HCC. CTLA-4 receptors are constitutively expressed on Tregs, and expression on effector CD4+ and CD8+ T cells is induced following activation[73]. Once expressed, CTLA-4 outcompetes the costimulatory receptor CD28 for binding to CD80 and CD86 on antigen-presenting cells (APCs)[74]. In the TME, CTLA-4 is expressed on infiltrating Tregs, promoting immune escape[73]. Dual blockade of PD-1 and CTLA-4 in preclinical models led to effector T cell upregulation and inhibitor T cell downregulation[75]. Additionally, CTLA-4 inhibition has been shown to increase the effector T-cell to MDSC ratios within tumors, potentially overcoming primary resistance to anti-PD-1 monotherapy, associated with the presence of MDSCs and Tregs[76]. CTLA-4 inhibition may also play a role in durable immunotherapy responses, with proliferation of transition memory T-cells[77].”
Reviewer 2 Report
Comments and Suggestions for Authors
The authors discussed different immunotherapy combination in the treatment of hepatocellular carcinoma. They also highlighted challenges in identification of predictive biomarkers in hepatocellular carcinoma.
Suggestions:
- Please add explanation for durvalumab and tremelimumab, line 49- 50 and oxiplatin line 156, as you did for atezolizumab and bevacizumab in line 47
- Table 2. Number of patients is missing for CheckMate 459
- Please explain different findings in CheckMate-459 and CheckMate-040 (chapter 5. 5. Predictive Biomarkers in HCC: Challenges and Progress )
Author Response
Comment 1: Please add explanation for durvalumab and tremelimumab, line 49- 50 and oxiplatin line 156, as you did for atezolizumab and bevacizumab in line 47
Response to Comment 1: We appreciate the reviewer’s suggestion to further explain durvalumab, tremilimumab and oxaliplatin. We have added explanations for durvalumab and tremelimumab [Lines 54-55], similarly to how we described atezolizumab/ bevacizumab.
“Dual checkpoint blockade with durvalumab (an anti-PD-L1 antibody) and tremelimumab (an anti-CTLA-4 antibody)…”
We have also provided additional information about oxaliplatin, explaining its mechanism as an alkylating chemotherapy agent [Lines 173-174].
“In contrast, the KEYNOTE-394 trial in an Asian population of patients with advanced HCC who had progressed on or were intolerant to sorafenib or chemotherapy incorporating oxaliplatin (an alkylating agent)…”
These edits now offer a better understanding of mechanisms of action of these agents.
Comment 2: Table 2. Number of patients is missing for CheckMate 459
Response to Comment 2: We appreciate the reviewer’s attention to detail. We have added the number of patients for CheckMate-459 in the table to ensure completeness. Of note, based on other reviewer’s suggestion, we combined the original Tables 1 and 2 into a single table to consolidate all frontline trials into a single table. See now Table 1.
Comment 3: Please explain different findings in CheckMate-459 and CheckMate-040 (chapter 5. 5. Predictive Biomarkers in HCC: Challenges and Progress )
Response to Comment 3: We appreciate the reviewer’s feedback. More detailed findings of CheckMate-459 and CheckMate-040 are described earlier in the section on Development of Immunotherapy in HCC [Lines 152-165]:
“The CheckMate-040 study, a multicohort phase I-II trial of nivolumab in advanced HCC in both sorafenib naïve and experienced patients, was one of the key initial clinical studies to demonstrate a potential durable survival benefit in a proportion of patients, with responses seen regardless of HCC etiology or PD-L1 expression[30]. In this trial, nivolumab demonstrated an acceptable safety profile with response rate of up to 20% in the dose expansion cohort.
Numerous randomized phase III trials investigating single agent anti-PD-1 antibodies subsequently ensued. The CheckMate-459 trial, comparing nivolumab to sorafenib as first-line treatment in advanced HCC, was unable to show a significant improvement in OS of nivolumab (16.4 vs 14.7 months, HR 0.85, 95% CI 0.72-1.02), despite a higher response rate (15% vs 7%) and a favorable safety profile (Table 1). Possible reasons for the lack of OS benefit include the higher percentage of patients in the sorafenib arm receiving subsequent immunotherapy as well as time-varying hazard ratios with delayed separation of Kaplan-Meier curves[7].”
We have added additional details within the Predictive Biomarkers section as well, see Lines 380-395:
“Although PD-L1 expression has been utilized as a predictive biomarker for treatment effectiveness of anti-PD-(L)-1 therapy in other cancers, such as lung cancer[82] and gastroesophageal cancer[83], its prognostic and predictive value in HCC has been less conclusive. Some clinical trials such as CheckMate-459, the phase III study of nivolumab versus sorafenib as first-line treatment of advanced HCC, showed that positive tumor cell PD-L1 expression correlated with higher treatment response in patients receiving nivolumab[7]. In this study, patients with PD-L1 expression >1% demonstrated ORR of 28%, compared with ORR of 12% in patients with PD-L1 <1%. However, other studies such as CheckMate-040 which preceded CheckMate-459 and evaluated both nivolumab monotherapy and nivolumab plus ipilimumab combination therapy in different study cohorts, did not show a clear predictive effect of tumor cell PD-L1 expression on response. In its dose expansion phase of the nivolumab monotherapy cohort, the ORR was 26% in patients with PD-L1 expression >1% versus 19% in patients with PD-L1 <1%, suggesting response regardless of PD-L1 status[3,80]. Thus, there have been conflicting, inconsistent results regarding PD-L1, and ultimately unreliable predictive utility as a biomarker in HCC.”
Reviewer 3 Report
Comments and Suggestions for Authors
The manuscript provides a comprehensive review of immunotherapy advancements in hepatocellular carcinoma (HCC), with a focus on immune checkpoint inhibitors (ICIs) and their combinations with anti-angiogenic therapies. It presents important insights into clinical efficacy, but several areas require revision to enhance clarity and scientific rigor.
Major Revisions:
1) In-depth Discussion on Resistance Mechanisms: While the manuscript touches on the immune resistance mechanisms in HCC, a more detailed exploration of primary and acquired resistance mechanisms would be valuable. The review could benefit from more concrete examples of how these mechanisms impact the efficacy of combination therapies and how they may be addressed in future clinical settings. For example, a deeper analysis of how PD-1/PD-L1 resistance occurs in the context of T-cell exhaustion or changes in tumor antigen presentation would strengthen the manuscript's impact.
2) Predictive Biomarkers for Immunotherapy: The manuscript addresses the need for predictive biomarkers in HCC immunotherapy but lacks specific recommendations for their integration into clinical practice. The authors should provide a more detailed discussion on the current status of biomarkers like PD-L1 expression, T-cell infiltration, and immune signatures. A thorough exploration of how these biomarkers can be used to predict patient responses and guide treatment decisions is critical for advancing the clinical application of immunotherapy in HCC.
3) Clarity in the Presentation of Trial Results: The manuscript provides a thorough summary of various clinical trials, but the presentation of these results could be more concise and structured. The authors should consider summarizing key trial findings in a comparative table, highlighting the efficacy (OS, PFS, ORR) of different therapies to enhance clarity. Additionally, clearer explanations of how some trials failed to meet primary endpoints, such as those in the LEAP-002 and COSMIC-312 studies, would be helpful in understanding the limitations of current therapies.
4) Mechanisms of Immune Modulation and TME: While the manuscript briefly touches on the immune microenvironment (TME) in HCC, it would benefit from a more in-depth discussion of how different components of the TME (e.g., Tregs, MDSCs, TAMs) contribute to immune evasion and the efficacy of immunotherapies.
Minor Revisions:
1) Figures and Data Presentation: The manuscript includes important figures that effectively summarize the data, but some of the figures, particularly those related to survival analysis and immune responses, could benefit from more detailed captions. Specifically, the role of immune cells in tumor progression and response to therapy should be highlighted more clearly in the figure legends.
2) Comparison with Other Malignancies: The manuscript provides a solid foundation for immunotherapy in HCC, but it could benefit from a broader comparison with immunotherapy strategies in other solid tumors, such as melanoma and non-small cell lung cancer. This comparison would help contextualize the challenges and successes in HCC and provide insights into how strategies might be adapted or improved.
3) Ethical Considerations: The manuscript briefly mentions the use of immunotherapies in clinical trials but could include a discussion on ethical considerations, particularly regarding patient selection and informed consent for experimental therapies. Furthermore, ethical challenges in managing immune-related adverse events and their long-term implications for patients could be discussed.
4) Typographical and Grammatical Errors: There are minor typographical and grammatical issues that need to be addressed to improve the overall readability of the manuscript. For example, sentence structure in some sections could be simplified for better clarity, and certain terms (e.g., "immunosuppressive cell populations") could be more clearly defined.
Conclusion:
This manuscript provides an important contribution to the field of hepatocellular carcinoma immunotherapy. The authors have presented a comprehensive overview of the advancements in immune checkpoint therapy and combination approaches, along with the challenges associated with treatment resistance and biomarker identification. With the suggested revisions, particularly regarding the inclusion of more in-depth discussions on resistance mechanisms, biomarkers, and TME modulation, the manuscript will have a significant impact on guiding future research and clinical practice in the treatment of HCC.
Recommendation: Major revisions required, particularly in the areas of resistance mechanisms, biomarkers, and trial data presentation.
Comments on the Quality of English LanguageThe manuscript is well-written, with clear and precise language, though some sentences are complex and could be simplified for better clarity. Technical terms like PD-1, PD-L1, and CTLA-4 would benefit from brief explanations for accessibility. The structure is logical, but transitions between sections could be smoother. A few typographical errors and inconsistent abbreviation usage should be addressed. Figures and tables could use more detailed captions to clarify the data presented. Overall, the manuscript is strong but would benefit from minor revisions for clarity, readability, and consistency in terminology.
Author Response
Major Revisions:
Major Comment 1: In-depth Discussion on Resistance Mechanisms: While the manuscript touches on the immune resistance mechanisms in HCC, a more detailed exploration of primary and acquired resistance mechanisms would be valuable. The review could benefit from more concrete examples of how these mechanisms impact the efficacy of combination therapies and how they may be addressed in future clinical settings. For example, a deeper analysis of how PD-1/PD-L1 resistance occurs in the context of T-cell exhaustion or changes in tumor antigen presentation would strengthen the manuscript's impact.
Response to Major Comment 1: We appreciate the reviewer’s feedback to provide a more in-depth discussion on resistance mechanisms. We have expanded this section to provide more details on primary resistance (expanding upon intrinsic and extrinsic tumor related factors) and acquired resistance. See lines 179-257.
Major Comment 2: Predictive Biomarkers for Immunotherapy: The manuscript addresses the need for predictive biomarkers in HCC immunotherapy but lacks specific recommendations for their integration into clinical practice. The authors should provide a more detailed discussion on the current status of biomarkers like PD-L1 expression, T-cell infiltration, and immune signatures. A thorough exploration of how these biomarkers can be used to predict patient responses and guide treatment decisions is critical for advancing the clinical application of immunotherapy in HCC.
Response to Major Comment 2: We appreciate the reviewer’s suggestion to further explore how biomarkers studied to date may be used in current treatment decisions. We have expanded upon discussion of molecular biomarkers (PD-L1, TMB, MSI, POLE and POLD mutations) associated with immunotherapy response in other cancers and discuss their limitations in HCC. We more clearly state in the final thoughts of this section that despite what has been studied, there are unfortunately still no clear predictive biomarkers identified which can be incorporated into routine clinical practice. See lines 380-438.
Major Comment 3: Clarity in the Presentation of Trial Results: The manuscript provides a thorough summary of various clinical trials, but the presentation of these results could be more concise and structured. The authors should consider summarizing key trial findings in a comparative table, highlighting the efficacy (OS, PFS, ORR) of different therapies to enhance clarity. Additionally, clearer explanations of how some trials failed to meet primary endpoints, such as those in the LEAP-002 and COSMIC-312 studies, would be helpful in understanding the limitations of current therapies.
Response to Major Comment 3: We appreciate the reviewer’s suggestion and have altered Table 1 to incorporate all key frontline immunotherapy clinical trials to provide a more concise side-by-side comparison of trials that met their primary clinical endpoint and those that did not. In addition, we have added further discussion regarding why certain trials such as LEAP-002 and COSMIC-312 did not meet their primary clinical endpoints. See lines 302-325:
“In the COSMIC 312 study, cabozantinib plus atezolizumab did not demonstrate superiority in OS compared with sorafenib as first-line therapy in patients with advanced HCC (OS 15.4 vs 15.5 months, respectively; HR 0.90, 96% CI 0.69-1.18), but did improve PFS (6.8 vs 4.2 months; HR 0.63, 95% CI: 0.44-0.91)[68] (Table 1). Similarly the LEAP-002 study, which compared lenvatinib plus pembrolizumab versus lenvatinib was unable to reach its promising efficacy initially demonstrated in an earlier phase study[69]. This study did not demonstrate superiority in PFS (8.2 vs 8.0 months; HR 0.87, 95% CI:0.73-1.02) or OS (21.2 vs 19.0 months; HR 0.84, 95% CI 0.71-1.00) with the combination (Table 1). On the other hand, the CARES-310 trial, investigating camrelizumab (an anti-PD-1 antibody) plus rivoceranib (a VEGFR2-targeted TKI, also known as apatinib), demonstrated improved OS (22.1 vs 15.2 months; HR 0.62, 95% CI 0.49-0.80) and PFS (5.6 vs 3.7 months; HR 0.52, 95% CI: 0.41-0.65) (Table 1). However, 83% of patients in this study were notably enrolled from Asia; thus this combination has had limited adoption worldwide[70]. Reasons for the failures of COSMIC-312 and LEAP-002 to meet their primary endpoint of OS remain unclear. However, possible explanations include receipt of efficacious second-line therapeutic strategies in the control arm, including immunotherapy[71]. This may account for the superior PFS observed with atezolizumab/cabozantinib over sorafenib in COSMIC-312, but ultimately no OS benefit. Furthermore, underlying etiology has been postulated to explain differences in the survival outcomes. Exploratory subgroup analyses of COSMIC-312 showed that patients with hepatitis B derived clearer PFS and OS benefit than other etiology subgroups[72]. These findings may similarly raise the question in CARES-310 of whether the OS superiority of camrelizumab/rivoceranib in the overall study population may have been driven by the benefit derived in the predominantly Asian population with Hepatitis B etiology. “
Major Comment 4: Mechanisms of Immune Modulation and TME: While the manuscript briefly touches on the immune microenvironment (TME) in HCC, it would benefit from a more in-depth discussion of how different components of the TME (e.g., Tregs, MDSCs, TAMs) contribute to immune evasion and the efficacy of immunotherapies.
Response to Major Comment 4: We appreciate the reviewer’s suggestion. We have integrated discussion of how different components of the TME such as Tregs, MDSCs and TAMs contribute to immune evasion in the expanded discussion of mechanisms of resistance to immunotherapy. See lines 179-257.
Minor Revisions:
Minor Comment 1: Figures and Data Presentation: The manuscript includes important figures that effectively summarize the data, but some of the figures, particularly those related to survival analysis and immune responses, could benefit from more detailed captions. Specifically, the role of immune cells in tumor progression and response to therapy should be highlighted more clearly in the figure legends.
Response to Minor Comment 1: We appreciate the reviewer’s suggestion. We have consolidated all clinical trial data into a single table (Table 1). We have also added further description to the Figure 1 caption.
Minor Comment 2: Comparison with Other Malignancies: The manuscript provides a solid foundation for immunotherapy in HCC, but it could benefit from a broader comparison with immunotherapy strategies in other solid tumors, such as melanoma and non-small cell lung cancer. This comparison would help contextualize the challenges and successes in HCC and provide insights into how strategies might be adapted or improved.
Response to Minor Comment 2: We appreciate the reviewer’s suggestion to provide a broader comparison of immunotherapy strategies in HCC with those in other solid tumors. We have renamed the section “Immunotherapy in HCC: Future Directions” to “Immunotherapy in HCC: Future Directions and Challenges”. We have added comparison to treatment strategies in melanoma and NSCLC, highlighting differences of immunotherapy integration into earlier stage disease, biomarker driven patient selection, and novel immune approaches. This has helped contextualize the current state of immunotherapy in HCC as compared to other malignancies and offers insights into potential future directions. See lines 510-530:
“Despite substantial progress, advances in immunotherapy in HCC has lagged behind other malignancies, such as melanoma and non-small cell lung cancer (NSCLC). One difference is the earlier integration of immunotherapy in melanoma and NSCLC, including the neoadjuvant and adjuvant settings. Preoperative and postoperative combination immunotherapy strategies have not yet been incorporated into standard practice in HCC. The randomized IMbrave-50 trial evaluating adjuvant atezolizumab/bevacizumab versus observation following resection or ablation in patients with “high-risk” HCC showed no improvement in relapse free survival (RFS) (median RFS 33.2 months with the combination vs 36.0 months with active surveillance, HR 0.90 [95% CI 0.72-1.12]) after extended follow-up, with mature OS data still pending[112]. However, more recent research has suggested that combined modality therapy for unresectable HCC without extrahepatic spread may improve clinical outcomes. In the EMERALD-1 study, patients with unresectable, liver-confined HCC amenable to transarterial chemoembolization (TACE) who received TACE plus durvalumab/ bevacizumab showed superior PFS compared with those treated with TACE alone (median PFS 15.0 vs 8.2 months, HR 0.77 (95% CI 0.61-0.98, p=0.032)[113]. Similarly, the LEAP-002 study investigating pembrolizumab/lenvatinib plus TACE versus TACE alone showed a significant improvement in PFS for the combination (median PFS 14.6 vs 10.0 months, HR 0.66 (95% CI 0.51-0.84, p=0.0002). However, further data is needed to clarify whether combined modality therapy in these studies ultimately improves the overall survival of patients.”
Minor Comment 3: Ethical Considerations: The manuscript briefly mentions the use of immunotherapies in clinical trials but could include a discussion on ethical considerations, particularly regarding patient selection and informed consent for experimental therapies. Furthermore, ethical challenges in managing immune-related adverse events and their long-term implications for patients could be discussed.
Response to Minor Comment 3: We appreciate the reviewer’s suggestion and have added further discussion regarding patient selection criteria in clinical studies. We highlight inclusion primarily of Child Pugh B patients in clinical trials, which impacts applicability of results to a broader population of patients with HCC, particularly those with decompensated liver function. Ultimately, there is a need to balance safety with equitable access for patients with more advanced liver disease. See Lines 527-538:
“Finally, beyond clinical efficacy, patient selection criteria in clinical trials for advanced HCC have raised concerns regarding the applicability of newer treatment options for all patients in the real-world setting. This is largely due to the inclusion of patients only with preserved liver function (i.e. Child-Pugh A) in most clinical studies, due to concerns regarding toxicity and tolerability with decompensated liver disease, where liver dysfunction may be a competing factor in a patient’s morbidity and mortality. Data supporting the utilization of ICI monotherapy in Child-Pugh B patients has been mainly derived from retrospective studies, and robust treatment guidelines for this population are lacking. While these studies have generally shown overall tolerability to treatment, outcomes are often less favorable[114]. Ongoing studies are being conducted to evaluate the safety and efficacy of combination immunotherapy strategies such as atezolizumab/bevacizumab in patients with advanced HCC and Child-Pugh B cirrhosis[115]. Future trials must balance patient selection with inclusive access to treatment, particularly in patients with impaired or borderline liver function who may still derive clinical benefit from novel therapies.”
Minor Comment 4: Typographical and Grammatical Errors: There are minor typographical and grammatical issues that need to be addressed to improve the overall readability of the manuscript. For example, sentence structure in some sections could be simplified for better clarity, and certain terms (e.g., "immunosuppressive cell populations") could be more clearly defined.
Response to Minor Comment 4: We appreciate the reviewer’s careful reading of our manuscript. We have thoroughly reviewed the text to address typographical and grammatical errors, improve sentence structure, and ensure terms are more clearly defined. These revisions will help improve readability.
Reviewer 4 Report
Comments and Suggestions for Authors
The manuscript “Advances in Immunotherapy in Hepatocellular Carcinoma” is a comprehensive review of clinical trials involving single and combined immunotherapies for the treatment of advanced liver cancer. The text is well-written and well-organized, containing relevant information for researchers working in this field. It effectively highlights new R&D opportunities that could enhance the success of future treatments for this disease.
As a single suggestion, I recommend providing a more detailed description in the legend of Figure 1 to help readers better understand all the information summarized and presented there.
Author Response
Comment 1: As a single suggestion, I recommend providing a more detailed description in the legend of Figure 1 to help readers better understand all the information summarized and presented there.
Response to Comment 1: We appreciate the reviewer’s suggestion to provide a more detailed description in the legend of Figure 1 to improve clarity for readers. We revised the legend to better describe the information presented, including categorization of treatments, line of treatment, and trial phases. This revision helps improve the figure’s clarity to readers.
Round 2
Reviewer 3 Report
Comments and Suggestions for Authors
n/a